# Viral Complexity

**DOI:** 10.3390/biom12081061

**Published:** 2022-07-30

**Authors:** Frank O. Aylward, Mohammad Moniruzzaman

**Affiliations:** 1Department of Biological Sciences, Virginia Tech, Blacksburg, VA 24061, USA; 2Center for Emerging, Zoonotic, and Arthropod-Borne Pathogens, Virginia Tech, Blacksburg, VA 24061, USA; 3Rosenstiel School of Marine and Atmospheric Science, University of Miami, Coral Gables, FL 33149, USA; monir@vt.edu

**Keywords:** viral diversity, giant viruses, jumbo bacteriophages, DNA viruses, virocell

## Abstract

Although traditionally viewed as streamlined and simple, discoveries over the last century have revealed that viruses can exhibit surprisingly complex physical structures, genomic organization, ecological interactions, and evolutionary histories. Viruses can have physical dimensions and genome lengths that exceed many cellular lineages, and their infection strategies can involve a remarkable level of physiological remodeling of their host cells. Virus–virus communication and widespread forms of hyperparasitism have been shown to be common in the virosphere, demonstrating that dynamic ecological interactions often shape their success. And the evolutionary histories of viruses are often fraught with complexities, with chimeric genomes including genes derived from numerous distinct sources or evolved de novo. Here we will discuss many aspects of this viral complexity, with particular emphasis on large DNA viruses, and provide an outlook for future research.

## 1. Introduction

Ever since their discovery by Dimitry Ivanovsky and Martinus Beijerinck, viruses have typically been viewed as diminutive biological entities that lack the physical, metabolic, ecological, and evolutionary complexity that characterizes cellular life. Indeed, many early molecular biologists were attracted to the study of viruses specifically because of this simplicity. For example, Max Delbrück and colleagues in the famous Phage Group undertook detailed studies of bacteriophage because they viewed them as fundamental biological entities and would therefore provide insight into the essential nature of genes and biological replication [1,2]. In many ways these expectations were borne out, and early work on bacteriophages did indeed lead to revolutionary discoveries that spurred the development of molecular biology. Yet over the last several decades this view of viral simplicity has also been contradicted with many startling discoveries that have highlighted remarkable examples of complexity in the virosphere. This has led to a gradual yet fundamental shift in our understanding of viruses.

Here we will provide a broad overview of recent developments that underscore the complex nature of viruses and provide a brief outlook on the future. We have divided up the different themes of viral complexity in terms of a general discussion of virion and genome size, viral infection strategies and virocell metabolism, viral ecology and virus–virus interactions, and the co-evolution and gene exchange between viruses and their hosts (Figure 1). These divisions are somewhat artificial and in many cases these areas of study are deeply intertwined, but this organization provides a starting point for assessing recent discoveries and emerging themes. In most sections we will focus on several groups of large DNA viruses—in particular “giant viruses” that infect eukaryotes (phylum *Nucleocytoviricota*) and “jumbo” bacteriophages (class *Caudoviricetes*)—for the reason that many aspects of viral complexity are most dramatically highlighted in these groups. However, many examples of unexpectedly complex ecological interactions and virus–host dynamics have been increasingly discovered across a wide range of viruses of different sizes, and we will therefore discuss some examples from smaller viruses as well. Lastly, we focus our attention primarily upon viruses of microbes because these represent by far the largest reservoir of viral diversity on Earth. 

Complexity is of course a notoriously difficult property to define and quantify, and some may question whether this is even a meaningful term to use in this context. It is certainly difficult to imagine a single definition that will suffice for a quantitative comparison of viral structure, genomics, ecology, and evolution. Despite this ambiguity, we have forged ahead with the use of this term because it is appropriate for a qualitative discussion of disparate phenomena where viruses have been shown to exhibit unexpectedly high levels of organization that are on par with, or in some cases exceeding, cellular life. Undoubtedly, discoveries in the future will continue to clarify the nature of these characteristics and provide deeper insight into their prevalence in the virosphere. 

## 2. Virion and Genome Size

The physical dimensions of virions are perhaps the most readily quantified traits of viruses that can be compared both between viruses and with cellular life. Early studies focusing on Tobacco Mosaic Virus (TMV) referred to viruses as “contagium vivum fluidum” (contagious living fluid) due to its diminutive size relative to cells and ability to pass through the Chamberland filter commonly used in early microbiology studies [5]. This was one of the earliest clues that viruses were unique biological entities that were distinct from cells, which was confirmed with later structural studies in the 1930s and 1940s. This view of viruses as small, subcellular entities persisted for many decades until studies of large DNA viruses revealed that many virions can reach much larger sizes. Most notable are eukaryotic viruses of the phylum *Nucleocytoviricota*, of which the poxviruses are the most well-known. *Variola major*, the causative agent of smallpox, has been known to humanity for thousands of years in terms of the deadly impact on civilization, although its identity as a virus was not discovered until much later [6]. The large size of *V. major* virions allowed for it to be studied in the late 1800s—much earlier than other viral groups—using light microscopy and staining-based approaches that were developed for pathogenic bacteria [7]. Moreover, the close relative of *V. major*, Vaccinia virus, was used by Edward Jenner to develop the first vaccine in the 1700s, well before the distinction between viruses and cells was known [8]. Thus, even though research in poxviruses has a long history that predates that of other viruses, their large size obscured differences with cells. Many of the hallmark discoveries into the unique biological properties of viruses were therefore made with smaller viruses, such as TMV, where physical differences with cells are more pronounced. 

In addition to the poxviruses, the *Nucleocytoviricota* include several other families that were discovered in the 20th century. Members of the *Ascoviridae, Iridoviridae, Phycodnaviridae*, and *Asfarviridae* were first described in the 1900s, though the evolutionary links between these families remained largely enigmatic until comparative genomic studies in the early 2000s revealed their common evolutionary origins [9,10]. African Swine Fever Virus (ASFV), until recently the only cultivated member of the *Asfarviridae*, is an emerging swine pathogen first identified in Africa in the early 1900s [11], while the *Iridoviridae* and *Ascoviridae* are closely related lineages that infect a wide range of insects, amphibians, and fish that were first described in the mid-20th century [12]. Together with the poxviruses, these viral groups have icosahedral capsids ~100–200 nm in size and genomes up to 200 kbp in length. Chloroviruses in the family *Phycodnaviridae* were first isolated from a symbiotic species of *Chlorella* associated with *Paramecium bursaria* [13]; these viruses also have capsid sizes around ~190 nm in diameter, but their genomes were subsequently found to be larger than other dsDNA viruses of eukaryotes (>300 kbp) [14,15]. The large physical dimensions and genome length of the chloroviruses led some to refer to them as “giant viruses” starting in the 1990s [16,17]. 

Large viruses other than those in the *Nucleocytoviricota* were also examined in-depth starting in the 20th century. For example, in 1968 a large bacteriophage referred to simply as “phage G” was isolated from a bacillus species (the host was initially referred to as *Bacillus megaterium* but was subsequently found to be a member of the genus *Lysinabacillus* [18]). This phage has a 180 nm capsid diameter and 450 nm total length, making it markedly larger than the 300 × 18 nm size of TMV [18,19]. Moreover, phage G also encodes a genome that is nearly 500 kbp in length, which is remarkably large for a bacteriophage and remains the largest among cultivated tailed phages. Other large bacteriophages were described subsequently and became important experimental systems, such as *Pseudomonas* phage φKZ, which was isolated in 1978 and also found to have notably large physical dimensions and genome size (120 nm capsid, 180 nm tail length, 280 kbp genome) [20,21]. These phages are now colloquially referred to as “jumbo bacteriophages”, although this term technically refers to their large genomes (>200 kbp) rather than their physical dimensions per se. Numerous jumbo phages have been isolated in recent years, likely due to the upsurge in phage isolation that has occurred because of the success of the popular Sea-Phages course that integrates phage isolation into undergraduate education [22], as well as the renewed interest in phage therapy to combat antibiotic resistant bacterial infections. Although still a minority compared to smaller phages, over 300 jumbo phages have been cultivated and had their genomes sequenced [23]. Moreover, cultivation-independent approaches have reconstructed genomes of many more of these phages, some with genome sizes >700 kbp, and suggested that they are ubiquitous in the biosphere [24,25,26,27]. 

One last lineage of viruses with notably large virions and genomes that was first described in the 20th century are the herpesviruses. Although not reaching the same scale at the extremes of virion size and genome length as the *Nucleocytoviricota* and jumbo phages, herpesviruses still exhibit virions between 150–200 nm in diameter and genomes up to ~240 kbp in length. Herpesviruses have a HK97-fold capsid and are therefore classified within the realm *Duplodnaviria*, and it is likely that they share an ancient evolutionary link with tailed bacteriophages [28]. Similar to poxviruses, herpesviruses were known for their ill effects on human health long before their classification as viruses became clear; as a result, the history of their epidemiology is far older than that of their virion morphology or genomics. Although long considered to be vertebrate pathogens, some species that infect molluscs were isolated starting in the early 2000s [29,30,31], and metagenomic approaches have identified numerous others in environmental samples [32]. 

Our understanding of viral complexity was transformed in the early 2000s with the discovery of *Acanthamoeba polyphaga* mimivirus, a virus with 750 nm diameter and a 1.2 Mbp genome [33,34]. The complexity of mimivirus was shocking both in terms of its physical dimensions as well as its genome size: Light microscopy-based studies had previously mistaken mimivirus particles for cells, and at the time the mimivirus genome was over twice as large as the smallest cellular genome. These revelations irrevocably altered the traditional view of viruses as simple “filterable infectious agents” [35]. The complexity of mimivirus is so great that it led to the initial hypothesis that it may be a remnant of an ancient cellular lineage, although subsequent phylogenetic analysis revealed clear placement within the *Nucleocytoviricota* and a likely emergence from smaller DNA viruses [36,37]. Genomic analysis revealed a large number of genes never before observed in viral genomes, including several tRNA synthetases, translation initiation factors, chaperones, and genes involved in amino acid metabolism, substantially expanding the scope of viral-encoded functions known at that time. Indeed, the prefix “mimi”—derived from its “mimicking microbe” appearance—was given to this virus due to its cell-like features, and in this case gigantism may have evolved as a mechanism to resemble cellular prey and thereby induce phagocytosis by their amoeba host [38,39,40].

Since the discovery of mimivirus, our understanding of viral complexity has expanded considerably with the discovery of numerous other giant viruses of eukaryotes. Various species of amoeba have been used as the host to cultivate a wide range of these viruses, including *Pithovirus sibericum*, pandoraviruses, and other relatives of mimivirus [41,42,43]. These findings have led to new records for both virion size (1.5 um for *P. sibericum*) and genome length (>2.5 Mbp for pandoraviruses). A variety of other giant viruses with complex genomes have also been isolated from other protist hosts, including green algae and flagellate protozoa [44,45,46,47,48]. Early cultivation-independent studies demonstrated that these viruses are surprisingly common in the biosphere [49,50,51,52], and several later studies analyzed metagenome-assembled genomes of giant viruses from a wide range of different ecosystems, revealing a vast phylogenetic breadth [50,53,54,55,56,57,58,59]. Further, detailed molecular experimentation has revealed that the infection strategies of these viruses can differ markedly, underscoring the varied virus–host interactions that have evolved in this group. For example, while some strictly replicate in the cytoplasm, others have infection stages that take place in the nucleus [60].

## 3. Infection Strategy, Virocell Metabolism, and Viral Structures

Recent work on large DNA viruses has substantially broadened the scope of viral infection strategies, raising questions regarding the full extent to which viruses manipulate the physiology of their hosts during infection. Indeed, the complex functional repertoires encoded in the genomes of giant viruses have led some to describe these viruses as “quasi-autonomous” from their hosts [61,62]. The virocell concept is a key organizing principle in virology that is useful for understanding cellular physiological shifts that take place during infection [63]. The virocell concept emphasizes the intracellular activities of viruses during infection over their extracellular phase, and thereby promotes the view of viruses as dynamic biological agents with their own form of metabolism (i.e., cellular metabolism during infection). Viral manipulation of host physiology is hardly limited to large DNA viruses; in fact, numerous studies have elucidated how smaller viruses of plants and animals hijack and rewire critical host processes to ramp up virus production [64]. The virocell concept takes on a new dimension in the context of large DNA viruses, however, due to their large genomes and particularly complex viral-encoded strategies for cellular manipulation. For example, the manipulation of cellular central carbon metabolism and cytoskeletal networks during infection have been widely reported in a diverse array of viruses, but some members of the *Nucleocytoviricota* stand out in that they encode their own copies of glycolysis, TCA cycle, actin, and myosin genes in their genomes. 

The full list of cell-like functions that are encoded in giant viruses is now long and impressive; it includes genes involved in glycolysis [53], the TCA cycle [53,65], fermentation [46], the cytoskeleton [66,67,68], DNA packaging (histones) [69,70], light sensing (rhodopsins and chlorophyll binding proteins), sphingolipid metabolism [71,72], translation [34,73,74], glycosyl transferases [75], nutrient and ion transport [53,76,77], and more. Many of these genes have been discovered only recently, and their precise role in infection remains unclear. Functional predictions alone must be treated with some caution owing to the propensity of viruses to co-opt proteins for alternative functions. Two excellent examples that have been discovered recently include a glycosyl hydrolase and oxidoreductase that have been co-opted to function as structural proteins in pandoraviruses and mimiviruses, respectively [78,79]. At the same time, other proteins may have retained similar functions to their cellular homologs despite diverging so far that sequence homology is no longer detectable [80]. Immunomodulatory genes that are commonly encoded in poxviruses, asfarviruses, and herpesviruses are excellent examples of this. Nonetheless, this vast array of “cell-like” functions underscores the diverse mechanisms employed by large DNA viruses for takeover of their cellular hosts during infection. 

Relatively less is known about the functional repertoires of jumbo phages owing to the lack of detectable homologs in many genomes and the relatively recent discovery of many new groups [24,25,27,81]. Moreover, given the small number of jumbo phage genomes currently available, it is at present unclear if these viruses have any distinct functional capabilities that distinguish them from their smaller relatives. It is possible that some capabilities, such as CRISPR arrays and genes involved in the formation of anti-CRISPR proteinaceous shells, are more common in members of the *Caudoviricetes* that have particularly large genomes [24,82,83,84]. Moreover, several studies have noted multi-subunit RNA polymerases in jumbo phages—sometimes more than one—and it is possible that this offers a degree of transcriptional independence from the host that is more common in larger phages [85,86,87,88]. Aside from these, jumbo phages also encode a wide array of auxiliary metabolic genes (AMGs) involved in the manipulation of the host during infection [89], though many of these are shared with smaller relatives. Regardless of genome size, the diverse complement of phage-encoded AMGs is a fascinating element of viral complexity in its own right. Some of the best studied AMGs are photosystem components and genes involved in central carbon metabolism that are used by marine cyanophages during infection [90,91,92,93,94], but a wide array of AMGs involved in other metabolic processes have also been discovered [95,96]. 

In some large DNA viruses an important aspect of infection involves the packaging of numerous proteins into the virion. This is quite common in giant viruses, and in pandoraviruses and pithoviruses almost two hundred proteins have been detected in virions [41,97]. These proteins play a variety of roles; while various capsid and structural proteins are clearly involved in forming the virion itself, various enzymes involved in transcription, translation, protein processing, and resistance to redox stress are also included [98]. These typically include DNA and RNA polymerase subunits, ribonucleotide reductase, an mRNA capping enzyme, superoxide dismutase, and a variety of proteins with no clear functional prediction [98,99]. Given the quasi-autonomous self-replication of these viruses in the host cytoplasm, many of these virion-packaged proteins play key roles in the early stages of infection. Interestingly, RNA polymerase subunits have also been identified in the capsids of some jumbo phages, indicating this phenomenon is not unique to eukaryotic viruses [100,101]. Some recent studies have discovered unexpected proteins that are packaged in the virions of other giant viruses; one study identified enzymes with homology to TCA cycle components in pandoraviruses (putative homologs of α-ketoglutarate decarboxylase and acetyl-coenzyme A synthetase) [102], while viral-encoded histones were found to be present in marseillevirus virions [69]. 

Within the *Nucleocytoviricota,* virions themselves also contain an inner membrane, and in at least some cases it has been shown that a transmembrane potential can be detected in free virions [102]. In the well-studied PBCV-1 this membrane potential has been shown to play a key role in the early stages of infection; through fusion of the membrane with the outer membrane of the host, the host membrane becomes depolarized, generating a force that propels the inner contents of the virion into the cell [77,103]. Viral potassium channels that are embedded in the viral membrane play key roles in this process. Recently membrane potential was detected in pandoraviruses, and it was suggested that the TCA cycle enzymes packaged in the virion of this virus play a role in maintaining this electrochemical gradient [102]. The prevalence of this membrane potential and its different roles across members of the *Nucleocytoviricota* are unclear, but it remains a tantalizing aspect of virion bioenergetics that will be important for future studies to examine. 

Another recent finding that challenges the view of virions as inert particles involves the ability of some chloroviruses to manipulate the chemotactic behavior of cellular organisms, in effect “luring” prey. This fascinating example involves the phagotrophic ciliate *Paramecium bursaria,* which harbors endosymbiont *Chlorella* algae that are frequently infected by giant viruses (chloroviruses) in natural settings. *Chlorella* cells within *P. bursaria* are protected from virus attack, but predation of *P. bursaria* by copepods or other protists releases susceptible *Chlorella* cells [104,105]. Interestingly, chloroviruses often adhere to the outside of *P. bursaria* cells [106], and are therefore primed to infect their host algae once the latter are released from their ciliate host. Chlorovirus virions appear to contain a soluble compound that acts as a chemoattractant for *P. bursaria*, thereby luring the ciliate host such that virions can adhere to its membrane [107]; this likely increases the encounter frequency of chloroviruses and *Chlorella,* in the event that *Chlorella* cells are released from the *P. bursaria* cells as a consequence of predation. The exact mechanism of the chemoattraction of *P. bursaria* to chloroviruses is unclear, but evidence suggests it is a soluble compound that leaks from virions [107]. Altogether, these studies suggest that virions can use chemical signals to modulate the behavior of cellular organisms to promote their own propagation. If similar mechanisms are employed by other viruses they would certainly have broad implications for widespread ecological dynamics. 

Aside from the virion itself, many large DNA viruses form complex intracellular structures during infection. In giant viruses these virus factories, sometimes called viroplasms, are typically perinuclear membrane-associated structures in which virions are assembled and packaged. Their appearance has often been likened to that of a nucleus itself, raising important questions regarding whether viruses may have played a role in the early evolution of the nucleus in eukaryotes [108,109]. Virus factories formed by giant viruses can be quite large—several microns in diameter, in some cases [110,111]—but at least superficially similar structures are formed by a wide variety of other viruses during infection, including RNA viruses and jumbo bacteriophages [83,112]. Among phages, recent work on large *Pseudomonas* and *Serratia* viruses has shown that a tubulin homolog appears to play a role in the formation of these proteinaceous “phage nuclei” during the infection, and that these can play a role in anti-CRISPR defense [82,83,84,113]. Although structurally distinct from the virus factories found in the *Nucleocytoviricota*, phage nuclei appear to play a similar role in partitioning different enzymatic activities into different compartments during infection. 

## 4. Ecological Complexity

Research over the last few decades has made it clear that viruses also have unexpectedly complex ecological interactions—with hosts, other viruses, and selfish genetic elements—that further defy the paradigm of simplicity. The discovery of virophages is one of the most compelling; these viruses specifically parasitize giant viruses within the *Nucleocytoviricota* [114]. Although first characterized in mimiviruses, they have since been found in several relatives [115,116,117]. There has been some debate regarding whether virophage simply represent another group of satellite viruses, i.e., viruses that lack the ability to replicate on their own, and require a “helper” virus to complete their infection cycle [118]. A compelling argument can be made that virophages are fully functional viruses complete with the machinery needed for DNA polymerization and morphogenesis, however, and that they merely do so inside the virus factories of their host virus rather than independently in host cells [119]. This is an important distinction, because it implies that: (i) some members of the *Nucleocytoviricota* are so large that they can support their own bona fide viruses; and (ii) many protists can house a nested system of distinct viruses with their own complex ecological dynamics. Adding to this complexity, virophages can integrate into the genomes of the cellular host and re-activate upon infection by a giant virus, in effect producing an inducible antiviral defense [120]. Endogenous virophages have been found in the genomes of several protists, indicating this is a widespread phenomenon [121,122]. Interestingly, it has been found that mimiviruses harbor a defense mechanism against some virophages that was proposed to be analogous to the CRISPR-Cas system of prokaryotes [123]. Silencing of the genes within this system was shown to restore virophage replication in virus factories [124]. This is a fascinating example of a viral-encoded defense mechanism that targets a hyperparasitic virus, though the analogy between this system and CRISPR-Cas has been debated [125]. 

The ecological dynamics between giant viruses and virophage are further complicated by the presence of other selfish genetic elements. Transpovirons, for example, are linear plasmid-like episomes that parasitize both giant viruses and their virophage [126]. Like virophages, transpovirons also appear to replicate inside of virus factories, and can ultimately be found both inside of virions and integrated into the genomes of both giant virus and virophages [126]. Detailed analysis of virophage and transpovirons from different viral isolates identified complex ecological interactions between hosts and parasites [127]. Resident transpovirons can block the replication of other transpovirons that are introduced, and transpovirons appear better able to reproduce in viruses closely related to the virus from which they were isolated, indicating that complex eco-evolutionary feedbacks are at work. In another example of subviral parasitism, analysis of several genomes of a marine flagellate revealed the presence of numerous Ngaro superfamily retrotransposons that were associated with endogenous virophages [121]. The consequences of these retrotransposons for virophage proliferation remain unclear, but in some cases they may interrupt and pseudogenize virophage genes. Altogether, these examples indicate that “hyperparasitism” is frequent in the virosphere and can have important consequences that shape infection outcomes and viral evolution.

Bacteriophages also have complex interactions with other selfish genetic elements. A classic early example is enterobacteria phage P4, which was isolated in the early 1960s and found to require co-infection with the helper phage P2 in order to complete a successful lytic cycle [128,129]. Although P4 is capable of DNA replication and lysogenization, it requires the structural proteins of P2 or a related phage for morphogenesis and packaging. These co-infections necessitate complex cross-talk between the transcriptional programs of both phages, and can therefore be viewed as a form of intracellular ecological interaction. Since this discovery, a wide variety of other phage satellites have been discovered. Perhaps the best studied are the Phage Inducible Chromosomal Islands (PICIs), which, upon infection by a helper phage, can be induced to excise, replicate, and become encapsidated by the virions of their helper phages [130]. The PICIs can then spread between bacteria through transduction. PICIs often encode virulence factors, and have therefore been studied intensely for their role in the dissemination of these genes.

One area of research that has recently garnered intense interest has been the interplay between phages and host-encoded antiphage defenses [131]. A variety of new antiphage defenses have recently been discovered [132,133], and it is likely that the turnover of these elements is an important role in shaping patterns of microdiversity seen in natural bacterial populations [134]. These interactions can be surprisingly complex owing to the presence of some antiphage defenses on plasmids or other mobile genetic elements as well as the ability of some phages to co-opt these defenses for their own proliferation [135]. One remarkable example is the CRISPR/Cas system encoded by a *Vibrio* phage that targets and destroys an antiphage defense encoded by a host-encoded PICI-like element [136]. A variety of other sophisticated phage-encoded mechanisms for obfuscating host defenses have also been discovered, including the previously mentioned phage nucleus, highlighting how phage–host arms races can drive evolution in unexpected directions [82,137,138,139]. 

Many viruses in the environment can also have unexpected temporal variation in their activity, further complicating their ecological dynamics. This is best exemplified by marine viruses that show diel cycling in their activity, which is likely a sign of infections that vary together with the metabolic rhythms of the host. So far several studies have identified diel patterns in the transcriptional activity of marine viruses in the surface waters of different locations [67,140,141,142,143], suggesting that these patterns are widespread in the environment. The mechanisms that underpin this temporal partitioning of viral activity remain unclear, but a wide range of marine microbes undergo dramatic physiological changes throughout a diel cycle [144,145], and it is likely that this is reflected in a diel variation in receptor availability that mediates viral attachment. Diel partitioning of viral activity has important consequences for host–virus interactions and biogeochemical cycling, and further work will be necessary to disentangle the mechanistic details of these processes. 

Lastly, communication between viruses is an important phenomenon that can shape infection outcomes. There are several mechanisms of phage communication that have been reported, the earliest described being lysis inhibition [146,147,148]. During lysis inhibition, phage adsorption to an already-infected cell serves as a signal to lengthen the latent period, leading to larger burst sizes. High levels of secondary phage adsorption can be viewed as a signal that few uninfected cells are present in the immediate environment and that prompt lysis would be unfavorable, and lysis inhibition has therefore been hypothesized to be a mechanism to avoid lysis in an already phage-saturated environment [147]. More recently, other mechanisms of phage–phage communication involving small molecules have also been reported, the most recent involving a small peptide that is used to guide the switch from lytic to lysogenic infection in temperate phages [149]. Through this mechanism phages are able to assess the level of recent infections, which provides indirect information on the most favorable infection route. The peptides, referred to as arbitrium signals, were found to be phage-specific in many cases, presumably limiting cross-talk between the systems of different infecting phages. Subsequent studies went on to show that prophages continue to communicate via the arbitrium system after integration and that both the entry and exit from lysogeny are influenced by these dynamics [150]. Interestingly, both arbitrium-mediated communication and lysis inhibition can be viewed as signals sent from previously infected to currently infected cells that can be used to assess extracellular conditions and guide phage infection strategy. Similarities between these communication systems and bacterial quorum sensing have been noted, suggesting that there may be parallels between the social interactions of viruses and bacteria. 

The myriad interactions between viruses have led some to propose the establishment of a new field of “sociovirology” to explicitly examine the ecological and evolutionary consequences of these dynamics [151]. Complex eco-evolutionary dynamics stemming from the emergence of cheaters can occur in systems of a single virus infecting a single host [152], demonstrating that these considerations are inevitable in even relatively simple systems. In many cases cooperation between infecting viruses during high multiplicity infections can have both benefits and costs that have important consequences for infection outcome, further complicating the study of these phenomena [153]. It is likely that virus–virus interactions are ubiquitous in nature [154], underscoring the importance of establishing a rigorous framework for evaluating their dynamics.

## 5. Evolutionary Complexity

Viruses are a collection of disparate lineages with distinct evolutionary origins [155]. The emergence of different groups of viruses are often linked together through the fusion and rearrangement of genomic modules of different viral groups, however, which has been described as the “ultimate modularity” [156]. The evolution of many lineages of large DNA viruses is further complicated by their frequent acquisition of genes from cellular lineages, leading to chimeric genomes with genes derived from multiple distinct sources. The result is a complex palimpsest of gene exchange that has occurred to varying degrees and over varying breadths over long time periods [157]. 

The origin of *Nucleocytoviricota* likely dates back to the early diversification of eukaryotes; although early work on some giant viruses suggested that their large genomic repertoires are a consequence of their common descent from a fourth domain of cellular life, subsequent phylogenomic analyses confirmed that the *Nucleocytoviricota* emerged from smaller viruses and underwent subsequent periods of genomic expansion [36]. Analyses focusing on the Major Capsid Protein and packaging ATPase have been particularly instructive when examining viral origins because these essential viral genes were likely present in the earliest of these viruses. Both the MCP and ATPase of the *Nucleocytoviricota* have homologs in other smaller dsDNA viruses, notably virophage and polintoviruses, and it is therefore likely that these three groups share ancient evolutionary origins [158,159]. Indeed, due to these evolutionary links these viral groups are all classified within the kingdom *Bamfordvirae* in the recently adopted viral taxonomy [155].

The evolutionary origins of jumbo bacteriophages are less clear owing to a lack of conserved marker genes that can be used to infer phylogenetic relationships. Because the 200 kbp cutoff that is usually used to define jumbo bacteriophages is somewhat arbitrary, it is reasonable to assume that clades of jumbo phages will form groups with smaller viruses within the *Caudoviricetes*. Gene clustering analyses have generally borne out this expectation, though the higher-order evolutionary relationships between clades of jumbo phages remain unclear [25,160]. Large DNA viruses in the order *Herpesvirales* have a shared evolutionary history with tailed bacteriophages owing to their use of a homologous HK97-fold capsid, and both of these orders are currently classified in the kingdom *Heunggongvirae* [155]. Herpesvirus genomes do not exhibit the same dramatic level of genomic diversity that is found in other lineages of large DNA viruses [157], but they still show signatures of extensive gene acquisition from their hosts, in particular with regards to immunomodulatory genes [161]. Given the collective immensity of the host range of the *Caudoviricetes* and *Herpesvirales*, it is remarkable to consider that they likely share a common primordial progenitor [28]. 

The genomes of giant viruses, jumbo phages, and herpesviruses are all characterized by varying levels of mosaicism whereby genes appear to have been acquired from a wide range of sources, including cellular domains and other viruses. The genomes of large DNA viruses are therefore a chimeric assortment of genes acquired from diverse sources at different evolutionary periods [36,37,162]. The chimeric nature of giant virus genomes was recognized fairly early, with some authors even referring to mimivirus as “king of the gene robbers” due to its expanded complement of cellular-acquired genes [163]. Although it may seem perplexing how giant viruses have acquired genes from so many different sources, including from viruses that infect bacteria, many hosts of these viruses are heterotrophic protists that also feed on a wide assortment of bacteria and bacteriophages [164], while others house a wide range of bacterial and archaeal endosymbionts [165]. It is therefore possible that many protist hosts act as “melting pots” of genomic innovation in the sense that they house a varied assortment of bacteria, archaea, and viruses that potentially come into contact and exchange genes [166]. 

Viral genes with homologs in cellular lineages (i.e., virologs) have been acquired by giant viruses across a wide range of evolutionary timescales. Some of the earliest studies of virologs focused on the immunoregulatory genes present in poxviruses and noted that many of these genes were likely acquired by viruses in the early evolution of vertebrates [167,168]. The characteristic deep-branching placement of virologs has been noted many times in subsequent studies, including work focusing on viral histones [169,170], cytoskeletal components [66,67,171], glycolysis and TCA cycle components [53], and RNA polymerase subunits [172]. In these cases, genes were either acquired by viruses very early in the evolution of eukaryotes, perhaps even before the emergence of the last eukaryotic common ancestor (LECA), or the genes emerged first in viruses and were subsequently acquired by eukaryotes. Presently the best evidence for the viral origin of some virologs is found in viral-encoded actin (viractin), which appears to branch basal to eukaryotic homologs [68]. Many studies have identified virologs that have been transferred more recently, however, including transporters, rhodopsins, and sphingolipid metabolism genes where the donor eukaryotic lineages can be traced with some confidence [71,76,173]. The viral rhodopsins are interesting in that these genes have been acquired multiple times; some viral rhodopsins fall into a broad deep-branching clade indicating ancient HGT events [174], while others fall into a smaller clade nested into eukaryotic homologs, indicating more recent acquisition [173,175]. Collectively, these observations show that viral gene acquisition is a pervasive process that has occurred throughout the evolutionary history of these viruses.

The mosaic genomes of large DNA viruses are particularly perplexing because so few of the genes in these genomes have recognizable homologs in any sequence databases [27,81,160]. It therefore remains an enigma exactly how many of these genes arise. It is likely that some were acquired from cellular domains at some point and have diverged so far that sequence homology can no longer be detected. In support of this, one study of structural mimicry in viruses found that 70% of structural homologs (i.e., proteins with similar three-dimensional structures) that could be identified between viral and cellular proteins had no detectable sequence homology [80]. It is also likely that many viral genes evolve de novo in viruses, however, and some authors have argued that this is a major process in the evolution of novel protein families [176]. For example, numerous novel genes found in pandoraviruses have been postulated to arise through a process in which intergenic regions are expressed [42], and it is likely that similar processes occur in other viruses. Moreover, the evolution of novel protein families has been postulated to occur in RNA viruses through “overprinting”, a process in which proteins are produced through translation of alternative reading frames of existing genes, and similar processes may be at work in other viral groups [177]. Viral genomes are therefore a mosaic of genes acquired from cellular lineages and those that have been produced de novo, and rapid evolutionary rates often make it difficult to distinguish between these disparate processes. 

Given the long co-evolutionary history of eukaryotes and *Nucleocytoviricota*, it is unsurprising that eukaryotes have also acquired numerous genes from these viruses. Indeed, gene exchanges between giant viruses and eukaryotes occur in both directions, and several studies have noted that eukaryotic genomes contain genomic loci that derive from the endogenization of giant viruses. The first endogenous giant virus to be found in the genome of a eukaryote was reported in *Ectocarpus siliculosus*, a brown algae that is globally distributed in temperate coastal waters [178]. Virus-like particles associated with *E. siliculosus* were first observed in a New Zealand strain of this algae, but subsequent studies identified viruses in strains inhabiting a wide geographic range [179]. Various viruses with distinct morphologies have been observed in different populations of this alga, but *Ectocarpus siliculosus* virus-1 (EsV-1) has been studied in the most detail. This virus had been examined for several years before it was definitively shown that the viral genome integrated into that of the host brown algae [180], and subsequent genome sequencing of the host confirmed this observation [181]. This is a remarkable example of viral endogenization because it is closely linked to the complex life cycle of the host. Esv-1 Specifically infects the gametes of *E. siliculosus*, which lack a cell wall, and it subsequently integrates into the host genome. The viral DNA passes through subsequent cell divisions into all cells of the alga. The virus is propagated either through virion production, which takes place in the reproductive organs (gametangia and sporangia) and not vegetative cells, or via meiosis, in which endogenous viruses segregate into subsequent generations of gametes [179]. A strikingly similar virus–host interaction has been observed in a giant virus that infects related brown algae in the genus *Feldmannia*, suggesting that life-cycle dependent endogenization is common in other brown algae [182,183].

Subsequent studies have identified genes derived from giant viruses in a wide range of eukaryotic genomes, indicating that endogenization has occurred across the eukaryotic tree of life. Genes or genomic loci originating from giant viruses have been reported in amoeba, metazoa, plants, and several protist groups [184,185,186]. Due to difficulties obtaining high-quality genome assemblies for many eukaryotic lineages, it can sometimes be difficult to assess if viral sequences are indeed localized within host chromosomes, and it remains a possibility that some are episomal or signatures of a persistent infection. For example, the *Hydra vulgaris* genome assembly contains a ~400 kbp giant virus contig that cannot be clearly linked to a host chromosome [187]. In other cases, the genomic loci derived from giant viruses are short or highly fragmented and may represent ancient endogenization events that have subsequently undergone large-scale deletions or rearrangements. One elegant study provided an in-depth analysis of the bryophyte *Physcomitrella patens* and the lycophyte *Selaginella moellendorffii* and found evidence of many genomic loci derived from giant viruses, providing tantalizing evidence of associations between these viruses and the ancestors of modern plants [188]. Subsequent in-depth analysis of *P. patens* found evidence of siRNA production from some of these viral-derived loci, which the authors postulated may be a mechanism employed by gametes as an antiviral defense [189]. 

Although many of the genomic loci derived from giant viruses are short and contain only a few predicted genes, some eukaryotes harbor remarkably large segments of endogenous viral material. One study surveyed chlorophyte genomes and identified numerous Giant Endogenous Viral Elements (GEVEs) that reached sizes of up to almost 2 million base-pairs [190]. Over 15% of the chlorophyte genomes surveyed harbored GEVEs, and in some cases genomes harbored multiple distinct GEVEs that could be traced back to different viruses, underscoring the prevalence of endogenous giant viruses in some lineages. The most striking example was found in *Tetrabaena socialis*, where two GEVEs with a total length of 3.2 Mbp were present, and over 10% of total ORFs in the genome could be traced back to giant viruses. Further, although the initial genome of the model green algae *Chlamydomonas reinhardtii* did not contain a GEVE, several of these elements were found in other strains of the same species, suggesting that GEVEs contribute to intraspecies genomic variability [191]. In addition to GEVEs, smaller remnants of endogenous giant viruses have been identified in various chlorophyte genomes [190]. While the GEVEs represent recent endogenization events where deletions or large-scale chromosomal rearrangements have not yet obscured the contiguous viral regions, shorter discontiguous fragments with similarity to giant viruses likely represent remnants of ancient endogenization events. 

As for jumbo phages, it appears that current isolates are predominantly lytic, consistent with the larger genomes that are typically reported for virulent vs temperate phages [192]. It therefore seems likely that jumbo phages do not integrate into bacterial genomes as frequently as their smaller relatives, though the overall impact of phage integration into the genomes of their hosts is enormous. The “domestication” of prophages (i.e., integration followed by mutations that inactivate the prophage and prevent its future excision) appears to be a rampant phenomenon that has played a large role shaping bacterial genomes [193]. Indeed, the genomes of bacterial pathogens are enriched in prophages [194], and many key toxins and virulence factors are phage derived [195]. Some authors have even hypothesized that many toxins evolved in a form of symbiosis between lysogenic bacteriophage and their hosts to combat eukaryotic grazers [196]. A full description of the impact of bacteriophage integration on bacterial genomics is outside the scope of this review, but it is nonetheless clear that this is yet another aspect of the evolutionary complexity of phage–host interactions that has been recognized in the last few decades. 

## 6. Conclusions and Outlook

Viruses are a collection of diverse lineages of capsid-encoding genetic parasites with multiple distinct evolutionary origins [197,198]. Given their vast diversity and ancient evolutionary history, it is perhaps unsurprising that many have developed remarkably sophisticated infection strategies and ecological dynamics that belie the common notion of simplicity. Moreover, given their long coevolutionary history throughout their “great billion year war” with cellular life, there has been ample opportunity for gene exchange between viruses and cells that has dramatically shaped the evolutionary trajectories of both [199]. 

It is almost certain that further levels of complexity in the virosphere will be discovered in the near future. These will likely include the discovery of new viral lineages that break the current records for genome and virion sizes, new viral strategies for manipulating host metabolism during infection, novel antiviral defenses in cellular life and viral mechanisms for their evasion or co-option, unexpected mechanisms for virus–virus communication, and new insights into the impact that viruses have had on the evolution and diversification of cellular life. It is important to recognize that although isolation-based and cultivation-independent methods have dramatically expanded our view of viral diversity on Earth, most viral lineages have likely not yet been identified. For example, cultivation-independent methods such as metagenomics are typically successful at recovering only the most abundant lineages in a given sample, and as a result the “rare virosphere” remains an enormous reservoir of largely unexplored genetic diversity. At the same time, even relatively well-characterized viruses will continue to yield unexpected new insights as new aspects of their ecology and host interactions are examined in detail. Although the view of viruses as simple biological entities was useful in the early development of molecular biology, in the future it will be important to embrace complexity of the virosphere and search for the myriad viral structures, behaviors, and evolutionary dynamics that remain undiscovered. 

## Figures and Tables

**Figure 1 biomolecules-12-01061-f001:**
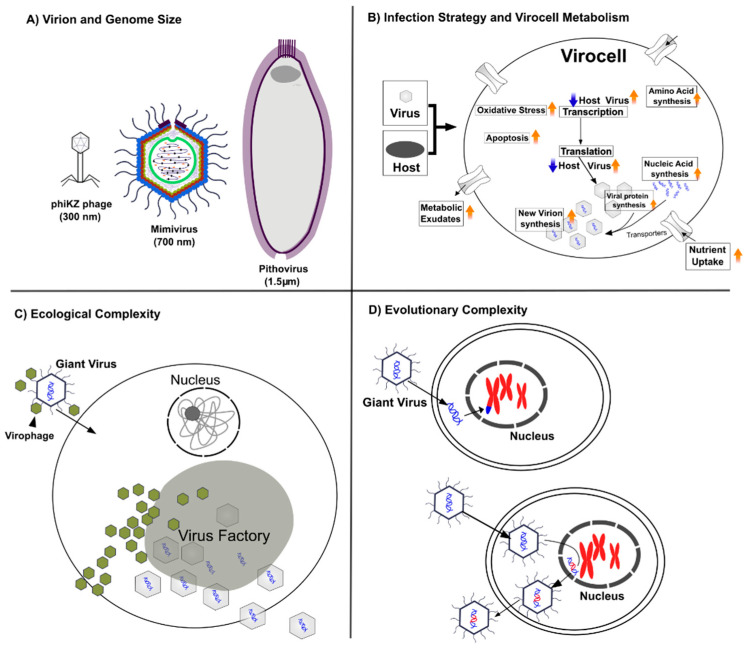
Major themes of viral complexity discussed here, together with illustrative examples. (**A**) Virion and genome size; some of the largest viruses discovered to date are given. (**B**) Virocell metabolism; example of virus-mediated physiological changes during infection (adapted from [3]). (**C**) Ecological complexity; giant virus–virophage interactions are shown. (**D**) Evolutionary complexity; gene exchange between viruses and hosts (adapted from [4]).

## Data Availability

Not applicable.

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
