# Peer review of "Viral Complexity"

_biomolecules, 2022, doi:10.3390/biom12081061_

Round 1
Reviewer 1 Report
This a very interesting review focusing on recent discoveries that transformed our vision of the virosphere by revealing an unexpected level of virus complexity. The review of the literature is exhaustive and very well done. The reading is easy and pleasant. I think it will be a hallmark review in the fields. I have only minor remarks.
The discussion about virophages could be slightly extended.
The discussion about the virocell concept could include a discussion about the implication of this concept for the origin of new genes.
Many evolutionists used to minimize the role of viruses in the creation of new genes because, focusing on the virion stage, they don’t understand how viruses can produce new genes. This is probably the origin of the “virus gene robber paradigm”. Indeed, the authors notice that “The chimeric nature of giant virus genomes was recognized fairly early, with some authors even referring to mimivirus as “king of the gene robbers” due to its expanded complement of cellular acquired gene (Moreira and Lopez-Garcia, 2005, also Moreira and Brochier-Armanet PMID: 18205905)”. However, these authors used to focus on genes with cellular origin that form only a small percentage (around 10%) of mimivirus genes. Most genes in giant viruses have no cellular homologs. In the framework of the virus gene robber paradigm, this was even interpreted by some authors, such as JM Claverie, as indicating that these genes came from extinct cells!!
It is easier to imagine that they simply originated in viruses. This is discussed by Forterre in PMID: 20551688 « Giant viruses: conflicts in revisiting the virus concept”.
The authors notice correctly that: “ It remains a possibility that many viral genes evolve de novo in viruses, however. For example, numerous novel genes found in pandoraviruses have been postulated to arise through a process in which intergenic regions are expressed, though it is unclear if a similar process may occur in jumbo phages”.
It is much likely that this process is not restricted to Pandoraviruses, neither to jumbo phage but that it is common to all viruses (and plasmids!), even small ones. For instance, the creation of new genes by small RNA viruses has been well documented (Rancurel et al., 2009, PMID: 19640978).
The possible formation of new genes in viral genomes by the same mechanisms that produces new genes in cellular genomes has been briefly discussed by Gaia and Forterre (PMID: 26894379). The authors could emphasize the fact that it appears important now to confirm these views by studying the mechanisms of de novo gene formation in viruses using similar strategies to those used to identify these mechanisms in cellular genomes.
Lane 18 in the summary, I will discuss should be We will discuss…
Lane 45 (order Caudovirales) better now to mention the class Caudoviricetes
Lane 563 : “Viruses are a collection of diverse lineages of selfish genetic parasites with multiple 563 distinct evolutionary origins, some of which may even predate cellular life [187,188]”.
I don’t like the idea that viruses may have even predate cellular life. This is confusing for me, since viruses, as we know them, cannot exist without proteins (ribosomes) and I cannot imagine ribosomes existing at a precellular stage.
Even if we define the first viruses simply as genetic RNA parasites, I also cannot imagine RNA replicons functioning in a precellular context, predating cellular life. I think it is not necessary having this sentence here. This would require discussing what one means by proto-cells, cells, modern cells etc…
Author Response
Reviewer 1
This a very interesting review focusing on recent discoveries that transformed our vision of the virosphere by revealing an unexpected level of virus complexity. The review of the literature is exhaustive and very well done. The reading is easy and pleasant. I think it will be a hallmark review in the fields. I have only minor remarks.
Thank you for your positive review and thorough comments.
The discussion about virophages could be slightly extended.
We agree that this is a fascinating topic, but given the manuscript is already longer than we intended we feel that it would be best if we kept this at the current length. For our purposes we felt it was necessary to discuss many different aspects of viral complexity, and as a consequence we cannot discuss any one topic in depth.
The discussion about the virocell concept could include a discussion about the implication of this concept for the origin of new genes.
We have expanded the discussion of new genes, but we put this in the section on genome evolution because we felt it was a better fit here (see below). Naturally genome evolution and virocell metabolism are intertwined, but for purposes of maintaining a cohesive narrative we believe the current organization is best.
Many evolutionists used to minimize the role of viruses in the creation of new genes because, focusing on the virion stage, they don’t understand how viruses can produce new genes. This is probably the origin of the “virus gene robber paradigm”. Indeed, the authors notice that “The chimeric nature of giant virus genomes was recognized fairly early, with some authors even referring to mimivirus as “king of the gene robbers” due to its expanded complement of cellular acquired gene (Moreira and Lopez-Garcia, 2005, also Moreira and Brochier-Armanet PMID: 18205905)”. However, these authors used to focus on genes with cellular origin that form only a small percentage (around 10%) of mimivirus genes. Most genes in giant viruses have no cellular homologs. In the framework of the virus gene robber paradigm, this was even interpreted by some authors, such as JM Claverie, as indicating that these genes came from extinct cells!!
It is easier to imagine that they simply originated in viruses. This is discussed by Forterre in PMID: 20551688 « Giant viruses: conflicts in revisiting the virus concept”.
The authors notice correctly that: “ It remains a possibility that many viral genes evolve de novo in viruses, however. For example, numerous novel genes found in pandoraviruses have been postulated to arise through a process in which intergenic regions are expressed, though it is unclear if a similar process may occur in jumbo phages”.
It is much likely that this process is not restricted to Pandoraviruses, neither to jumbo phage but that it is common to all viruses (and plasmids!), even small ones. For instance, the creation of new genes by small RNA viruses has been well documented (Rancurel et al., 2009, PMID: 19640978).
The possible formation of new genes in viral genomes by the same mechanisms that produces new genes in cellular genomes has been briefly discussed by Gaia and Forterre (PMID: 26894379). The authors could emphasize the fact that it appears important now to confirm these views by studying the mechanisms of de novo gene formation in viruses using similar strategies to those used to identify these mechanisms in cellular genomes.
Thank you for pointing this out. We agree this warrants more in-depth discussion, and we have broadened the focus of this paragraph so that it includes all viruses, not just jumbo phages. We also include the new citations and discuss how de novo evolution of new genes is a major force that shapes viral evolution. This is mentioned in the abstract as well.
Lane 18 in the summary, I will discuss should be We will discuss…
Corrected
Lane 45 (order Caudovirales) better now to mention the class Caudoviricetes
Changed
Lane 563 : “Viruses are a collection of diverse lineages of selfish genetic parasites with multiple 563 distinct evolutionary origins, some of which may even predate cellular life [187,188]”.
I don’t like the idea that viruses may have even predate cellular life. This is confusing for me, since viruses, as we know them, cannot exist without proteins (ribosomes) and I cannot imagine ribosomes existing at a precellular stage.
Even if we define the first viruses simply as genetic RNA parasites, I also cannot imagine RNA replicons functioning in a precellular context, predating cellular life. I think it is not necessary having this sentence here. This would require discussing what one means by proto-cells, cells, modern cells etc…
Thank you for pointing this out- we agree the logic here is confusing and we have deleted the second part of this sentence.
Reviewer 2 Report
The authors provide a useful overview of viral complexity, emphasizing developments in the study of giant viruses and their interactions with other viruses and their hosts. Minor edits are below.
L18 “Here [we] will discuss…” L324 “replicate [inside] of virus factories…” L336 “frequent in the [virosphere] and can have…” L340-344: check and fix “P2” versus “P4” L366-370: fix truncated sentence and paragraph L593: missing grant numberAuthor Response
The authors provide a useful overview of viral complexity, emphasizing developments in the study of giant viruses and their interactions with other viruses and their hosts. Minor edits are below.
L18 “Here [we] will discuss…” L324 “replicate [inside] of virus factories…” L336 “frequent in the [virosphere] and can have…” L340-344: check and fix “P2” versus “P4” L366-370: fix truncated sentence and paragraph L593: missing grant number
Thank you- these changes have been made.
Reviewer 3 Report
The manuscript entitled: Viral complexities discusses many aspects of fundamental virology complexities. The manuscript is detailed and comprehensive, but certain concerns need to be addressed to improve the quality of the manuscript.
Overall, the manuscript appears to be a compilation of the existing literature and lacks any opinion or ideas. In addition to compiling the important literature, authors should focus more on what is missing in the current literature and how this manuscript will help fill the missing knowledge gaps.
Line #36 mentions "...recent development" but the manuscript does not even mention the ongoing pandemic. Adding a few lines will be helpful.
Additionally, while describing the viral complexities, I strongly believe that authors should at least describe the viruses (in a few sentences) that have claimed most human lives, for example, Influenza, HIV, SARS, etc..
Manuscript also needs English editing. Some of the sentences are a bit exaggerated and non-scientific.
For instance, Line #563 selfish genetic parasite... ?
Line #18: "I" will discuss should be "we" will discuss.
Some references are incomplete: To list a few examples, Ref #47 lacks vol. Page no. etc and has a DOI number. Ref. 178 lacks any details.
Adding a table in each category summarizing the viruses and their characteristics described in the text will be very helpful for the readers.
Author Response
Reviewer 3
The manuscript entitled: Viral complexities discusses many aspects of fundamental virology complexities. The manuscript is detailed and comprehensive, but certain concerns need to be addressed to improve the quality of the manuscript.
Overall, the manuscript appears to be a compilation of the existing literature and lacks any opinion or ideas. In addition to compiling the important literature, authors should focus more on what is missing in the current literature and how this manuscript will help fill the missing knowledge gaps.
Thank you for your review. We intend this to be a review article, not an opinion piece, so we have omitted any controversial new perspectives. This is more of an article about how our perspectives of viruses have changed over the last 100 years, which is appropriate given the special issue topic.
Line #36 mentions "...recent development" but the manuscript does not even mention the ongoing pandemic. Adding a few lines will be helpful.
Additionally, while describing the viral complexities, I strongly believe that authors should at least describe the viruses (in a few sentences) that have claimed most human lives, for example, Influenza, HIV, SARS, etc..
The vast majority of viruses on Earth infect microbes, and from the perspective of viral diversity and evolution it therefore makes sense to prioritize these viruses rather than focusing on human pathogens. Moreover, the clearest examples of complexity are available are giant viruses and large phages, so we feel that integrating more text on human viruses, many of which are small RNA viruses, would be confusing and interrupt the flow of ideas. The literature on human pathogens is already quite vast, and we would like this review to focus on an alternative perspective. We have clarified in the second paragraph of the introduction that “...we focus our attention primarily upon viruses of microbes because these represent by far the largest reservoir of viral diversity on Earth”
Manuscript also needs English editing. Some of the sentences are a bit exaggerated and non-scientific.
For instance, Line #563 selfish genetic parasite... ?
Line #18: "I" will discuss should be "we" will discuss.
Some references are incomplete: To list a few examples, Ref #47 lacks vol. Page no. etc and has a DOI number. Ref. 178 lacks any details.
Thank you- we have adjusted these sentences and references. We note that viruses are often referred to as genetic parasites, so we are unsure why this would be inappropriate here, but we have nonetheless adjusted the language.
Adding a table in each category summarizing the viruses and their characteristics described in the text will be very helpful for the readers.
We feel that this would be confusing because we discuss a wide variety of different viruses in each of the sections, and when we mention individual viruses it is just to point out some specific examples that have been studied in detail. Providing a table would create the impression that the discussion applies to only a small number of well-characterized viruses, when in fact the trends we discuss are quite broadly represented in the virosphere. We realize our review article is somewhat atypical, but we feel that it fills an important gap in the literature.
Round 2
Reviewer 3 Report
The responses from the authors are satisfactory. The authors have made changes that have improved the quality of the manuscript. I recommend the acceptance of the manuscript.